# A High-Methanol-Permeation Resistivity Polyamide-Based Proton Exchange Membrane Fabricated via a Hyperbranching Design

**DOI:** 10.3390/polym16172480

**Published:** 2024-08-30

**Authors:** Liying Ma, Hongxia Song, Xiaofei Gong, Lu Chen, Jiangning Gong, Zhijiao Chen, Jing Shen, Manqi Gu

**Affiliations:** 1School of Chemistry and Materials Science, Guizhou Normal University, 116 Baoshan North Road, Guiyang 550001, China; 222100080287@gznu.edu.cn (H.S.); 201708045@gznu.edu.cn (L.C.); 100077896@gznu.edu.cn (J.G.); 201407055@gznu.edu.cn (Z.C.); 460141631@gznu.edu.cn (J.S.); 460106022@gznu.edu.cn (M.G.); 2Kaili No. 8 Middle School, 70 Qingjiang Road, Kaili 556000, China; 15185036586@163.com

**Keywords:** proton-exchange membrane, direct methanol fuel cells, methanol resistivity, blend membrane

## Abstract

Four non-fluorinated sulfonimide polyamides (s-PAs) were successfully synthesized and a series of membranes were prepared by blending s-PA with polyvinylidene fluoride (PVDF) to achieve high-methanol-permeation resistivity for direct methanol fuel cell (DMFC) applications. Four membranes were fabricated by blending 50 wt% PVDF with s-PA, named BPD-101, BPD-102, BPD-111 and BPD-211, respectively. The s-PA/PVDF membranes exhibit high methanol resistivity, especially for the BPD-111 membrane with methanol resistivity of 8.13 × 10^−7^ cm^2^/s, which is one order of magnitude smaller than that of the Nafion 117 membrane. The tensile strength of the BPD-111 membrane is 15 MPa, comparable to that of the Nafion 117 membrane. Moreover, the four membranes also show good thermal stability up to 230 °C. The BPD-*x* membrane exhibits good oxidative stability, and the measured residual weights of the BPD-111 membrane are 97% and 93% after treating in Fenton’s reagent (80 °C) for 1 h and 24 h, respectively. By considering the mechanical, thermal and dimensional properties, the polyamide proton-exchange membrane exhibits promising application potential for direct methanol fuel cells.

## 1. Introduction

The combustion of traditional fossil fuels generates harmful gases, such as NO, NO_2_, SO, SO_2_ and CO, giving rise to enormous environmental issues [1]. In recent years, proton exchange membrane fuel cells (PEMFCs), efficient and clean electrochemical devices for energy transformation, have become some of the most promising energy-converting devices to replace traditional fossil fuel power devices due to their high energy density and high efficiency [2,3,4,5,6,7]. Direct methanol fuel cells (DMFCs) have been regarded as some of the most promising portable electronic devices because of their convenience of design, extra-high energy density and the convenient storage/transportation of methanol [8,9,10,11,12,13]. However, wide range application of DMFC devices is hindered by the expensive noble metal catalyst [14,15] and severe methanol permeation [16,17] of proton exchange membranes (PEMs). PEMs play a pivotal role in DMFCs due to their prominent proton conductivity, effective methanol isolation, remarkable durability and excellent chemical stability. The methanol permeation to the cathode causes fuel wastage and leads to undesired mixed potential as well [18,19]. That is to say, methanol permeation reduces both the power output and energy density of DMFCs [20,21]. Up to now, perfluorosulfonic acid series membranes, such as commercial Nafion membranes (Dupont), are the most widely used PEMs due to their high electrochemical properties and long-term chemical stability [22,23,24,25]. But perfluorosulfonic acid series membranes are troublesome obstacles to the application of DMFCs due to their high cost and complex synthesis process, especially with regard to their serious methanol permeability [26,27,28]. Therefore, research on reducing methanol permeability and simplifying the synthesis process of PEMs is highly essential for DMFC application.

To date, a lot of polymeric proton conductors [29,30,31,32,33,34] that exhibit good performance in the ongoing PEM fuel cell market. Unfortunately, almost none of them can sustain the long-term chemical stability desired [35]. As an alternative to Nafion, countless strategies have been designed to improve methanol resistivity [36,37,38,39,40]. Sulfonated block copolymers [37] were successfully synthesized, and the thermal stability was stable up to 450 °C. A blended membrane based on SPEEK [39] was prepared at higher degrees of sulfonation of 80%, and the lowest permeability value of the blended membrane was 21 times lower than the Nafion 117 membrane.

Many sulfonated membranes have been widely investigated as potential alternative PEMs to be used in fuel cells, such as polyamide membranes [41,42], polyimide membranes [43,44], polyether ether ketone (PEEK) membranes [45], poly (arylene ether) membranes (PAE) [46] and poly (arylene ether ketone) (PAEK) membranes [47,48,49,50]. A poly(ether imide)-based membrane [51] was designed and synthesized that exhibits good chemical and mechanical stability. PAEK-based PEMs [52] that display excellent mechanical properties (tensile strength at break > 52 MPa) and good thermal stability (T_d5%_ > 259 °C) were synthesized. Sulfonated poly(ether sulfone) (SPES) [53] that exhibits high methanol resistivity was fabricated; the methanol permeability was as low as 3.05 × 10^−8^ cm^2^/S. PEEK-based composite membranes [54] that show low methanol permeability (3.19 × 10^−7^ cm^2^/S) were prepared.

In previous work, we synthesized sulfonated polyimide (s-PI) and the degree of branching was tertiary branching [44]. We designed and synthesized a sulfonimide polyamide (s-PA) with a quaternary branching-based blended membrane to overcome methanol crossover issues. Both works show good methanol resistivity and mechanical strength. Fluoride porous substrates, for example, polyvinylidene fluoride (PVDF), exhibit higher chemical and mechanical strength, arising from the strong C-F bond, than fluoride-free membranes [55]. Therefore, in order to fabricate mechanically strong PEMs, PVDF was employed as a binder due to its high flexibility, high chemical properties and great mechanical strength [56]. We designed and synthesized the s-PA by 1,2,4,5-benzenetetracarboxylic acid (BTA) and 2,5-diaminobenzenesulfonic acid (DSA) monomers, but the product exhibits high volume swelling because of their numerous hydrophilic sulfonic acid groups –SO_3_H and unreacted carboxylate groups (–COOH) resulting from steric hindrance. Therefore, a non-sulfonated hydrophobic monomer p-phenylenediamine (PPD) was introduced to regulate the hydrophobicity of the polymer. The structures of the polymers were adjusted by adding different ratios of PPD in synthesis. In addition, the formation of the connected network among the molecules is a consequence of hydrogen bones formed between the –COOH groups of molecules during the membrane preparation. Apparently, the connected network forms the basis for high mechanical strength and methanol resistivity. Of course, the IEC was also adjusted with the addition of PPD. The membranes were prepared by blending s-PA and PVDF, named BPD-101, BPD-102, BPD-111 and BPD-211, respectively. The main properties of blended PEMs, including their methanol permeability, proton conductivity, mechanical properties, water uptake, volume swelling, ion exchange capacity (IEC) and thermal stability, were systematically investigated and compared with the corresponding values of Nafion 117. The overall performances indicate that the s-PA/PVDF blended PEMs can potentially be applied in DMFCs.

## 2. Experimental

### 2.1. Materials

Polyvinylidene fluoride (PVDF, 98%), 1,2,4,5-benzenetetracarboxylic acid (BTA, 98%), 2,5-diaminobenzenesulfonic acid (DSA, 98%), p-phenylenediamine (PPD, 98%), triphenylphosphite (TPP, 99%) and N-methyl-2-pyrrolidone (NMP, 98%) were provided by Sigma-Aldrich Reagent, Ltd. (St. Louis, MO, USA). BTA, PVDF, DSA and PPD were dried under vacuum at 80 °C for 8 h. Dimethyl sulfoxide (DMSO, 98%), pyridine (Py, 98%), LiCl and NaOH were purchased from Sinopharm Chemical Reagent Co., Ltd. (Chengdu, China), and used as received without further purification. The Nafion 117 was purchased from Dupont Co. (Wilmington, DE, USA), and the Nafion 117 membrane was treated sequentially with 5% H_2_O_2_ solution, 1 M H_2_SO_4_ solution and deionized water at 100 °C for 1 h. TPP and NMP were used as received without any purification.

### 2.2. Synthesis

Following the synthetic routes shown in Figure 1, BTA (*x* = 2 mmol, 2 mmol, 2 mmol, 4 mmol), PPD (*y* = 0 mmol, 0 mmol, 2 mmol, 2 mmol), DSA (*z* = 2 mmol, 4 mmol, 2 mmol, 2 mmol, *x*:*y*:*z* = 1:0:1; 1:0:2; 1:1:1 and 2:1:1), TPP (1 mL), LiCl (0.32 g), Py (3 mL) and NMP (4 mL) were added into a 50 mL three-necked round-bottom flask, and the reaction was maintained at 100 °C for 20 h with stirring under an argon atmosphere. Then, the products were washed with methanol and deionized water several times and dried at 80 °C under vacuum for 24 h. The ratios of the monomers in the synthesis were 1:0:1, 1:0:2, 1:1:1 and 2:1:1, and the polymers were denoted as BPD-101, BPD-102, BPD-111 and BPD-211, respectively.

### 2.3. Membrane Preparation

A series of blended proton exchange membranes with 50 wt. % amounts of PVDF were fabricated via a solution-casting technique using dimethyl sulfoxide (DMSO) as the solvent. In 15 mL DMSO, 300 mg BPD/PVDF polymers was dissolved to form a homogeneous solution, then the solution was filtered and casted on a glass plate with a diameter of 7 cm, followed by drying in an oven at 80 °C for 24 h. All the dried proton exchange membranes were acidized in 1 M HCl solution for 24 h at ambient temperature, and then washed with ultrapure water several times to remove the HCl solution until the pH = 7. The membranes were denoted BPD-*x* (*x* is the ratio of monomers), e.g., BPD-101, BPD-102, BPD-111 and BPD-211. And the thickness of the membranes were measured to be 0.154 mm, 0.087 mm, 0.107 mm and 0.053 mm, respectively.

### 2.4. Characterizations and Measurements

^1^H NMR spectra of synthesized polymers were obtained on the 400 MHz NMR instrument (AVANCE III HD 400 MHz, Swiss BRUKER) using deuterated dimethyl sulfoxide (DMSO-d_6_) as solvent. Then, the polymer structure was recorded on an FT-IR Nicolet6700 spectrometer (Thermo Fisher Scientific, Waltham, MA, USA).

Thermogravimetric analyses (TGAs) of the proton exchange membrane samples were recorded with the NETZSCH simultaneous thermal analyzer (Netzsch, Waldkraiburg, Germany) with the heating rate of 10 °C/min from room temperature to 800 °C under nitrogen atmosphere. A 3 mg membrane was heated in a crucible to study the thermal behavior of the membrane.

Cross-section morphology of the proton exchange membranes was recorded using field emission scanning electron microscopy (SEM, SU8010, Hitachi, Tokyo, Japan). The atomic force microscopy (AFM) images of the proton exchange membranes were carried out using a Digital Instruments Multimode (SPM9700, Shimadzu, Kyoto, Japan). X-ray diffraction (XRD) spectra were measured using a Bruker AXS D8-Focus X-ray diffractometer (Karlsruhe, Germany). Molecular weights of the polymers were measured by Gel Permeation Chromatography (GPC) with Waters 2414 refractive index detector (Milford, UAS), using a 7 WAT-066224 column at 40 °C with N,N-Dimethylformamide (DMF) as mobile phase at a flow rate of 1.0 mL/min and polystyrene was used as standard.

The tensile stress and elongation at break of proton exchange membrane samples were determined by using a tensile machine Labthink XLW (PC) (Labthink, Jinan, China) with a 50 N load cell under the stretching speed of 25 mm/min at room temperature. The membrane samples were cut to rectangle size of 1 cm × 4 cm to study the mechanical properties of the membrane.

The proton exchange membrane was sheared into pieces. The dry weight was accurately weighed, and then the dry membrane was immersed into Fenton’s reagent (3 ppm FeSO_4_ in 3% H_2_O_2_) at 80 °C. The oxidative stability of the membrane was finally evaluated by the percentage of retained weight after the membrane was immersed in Fenton’s reagent for 1 and 24 h, respectively.

Proton conductivity (PC) of the proton exchange membrane sample was evaluated by using the four-electrode AC impedance method with temperature from 25 to 80 °C at full humidity, and the AC impedance parameter was adjusted between 0.1 MHz and 1 Hz. The proton conductivity test device is shown in Figure 1. The membrane (3 cm × 1 cm) was placed in the test cell to measure the proton conductivity. An electrochemical workstation CHI 604E (Shanghai, China) was used to evaluate proton conductive performances of the proton exchange membranes. The resistance of the proton exchange membrane was measured after the humidity and temperature were setup. The value of proton conductivity was calculated with the following equation:(1)$$\sigma =\frac{d}{Rtw}$$
where σ is the proton conductivity (S/cm), *d* represents the distance between the electrodes (cm) and *w* and *t* refer to the width (cm) and thickness (cm) of the membrane, respectively. *R* is the resistance received from the measured electro-chemical impedance spectroscopy (EIS) data (Ω).

Methanol permeability was obtained via a self-conceived H-type diaphragm diffusion device as show in Figure 2. The device has two chambers, one diffusion chamber filled with 10.0 mL 1 M methanol solution and the other sampling chamber filled with the same volume of deionized water. A proton exchange membrane was placed in the middle of the diffusion device. The chambers were separated by the membrane sample and stirred with a magnetic stirrer for diffusion at room temperature, which ensured the uniformity of the chamber during the test. The time interval of sampling was 1 h, and the whole sampling was completed within 6 h. The methanol concentration in the sampling chamber was measured via a gas chromatograph (9790II, Zhejiang Fuli Analytical Instruments Co., Ltd., Wenling, China). Methanol permeability (*p*, cm^2^/s) was calculated according to the following equation [57].
(2)$$p=\frac{l}{A}\times \frac{V_{B}}{C_{A}}\times \frac{\Delta C}{\Delta t}$$
where *p* is the methanol permeability (cm^2^/s), C_A_ represents the initial concentration of methanol (mol/cm^3^), V_B_ is the volume of the added solution (cm^3^) and A and l are the effective diffusion area (cm^2^) and the thickness (cm) of the membrane, respectively. ∆C stands for the methanol concentration (mol/cm^3^) and ∆t is the diffusion time (s).

Volume swelling (VS) and water uptake (WU) ratios of the proton exchange membrane samples were tested using the following process. Firstly, the sheared membrane sample was placed in deionized water for 24 h at room temperature, then the weight and the volume of the wet membrane were immediately measured after removal of the surface water with tissue paper. Subsequently, the membrane sample was dried at 80 °C for 24 h under reduced pressure to measure the weight and the volume of the dry membrane quickly. WU and VS ratios were obtained according to Equations (3) and (4):(3)$$\mathrm{WU}=\frac{\left(W_{\mathrm{Wet}}-W_{\mathrm{dry}}\right)}{W_{\mathrm{dry}}}\times 100\%$$
where W_dry_ and W_wet_ represent the weight of dry and wet membranes, respectively.
(4)$$\mathrm{VS}=\frac{V_{\mathrm{Wet}}-V_{\mathrm{dry}}}{V_{\mathrm{dry}}}\times 100\%$$
where V_wet_ and V_dry_ refer to the volume of the wet and dry membranes, respectively.

Ion exchange capacity (IEC, mmol/g) of the proton exchange membranes was estimated through classical acid-base titration. The weight of the membrane sample was recorded after drying completely. And then the membrane sample was immersed in 100 mL of 1 M NaCl solution for at least 24 h at ambient temperature so as to fully convert H^+^ to Na^+^. Finally, the contents of the H^+^ in the solution were titrated by 0.01 M standard NaOH solution using phenolphthalein as the indicator. The volume of the standard NaOH solution consumed was recorded. The operation was repeated three times to obtain exact data. The ion exchange capacity (IEC) value was calculated with Equation (5):(5)$$\mathrm{IEC}=\frac{V\times C}{W}\left(\mathrm{mmol}/g\right)$$
where C and V represent the concentration (mol/L) and the volume (mL) of the consumed standard NaOH solution, respectively. W is the weight of the dry membrane sample (g).

## 3. Results and Discussion

Successful polymerization of the BPD-*x* polymers with –COOH groups end-capped were verified with the ^1^H NMR spectra data in Figure 3. The monomer ratios of 1:0:1 and 2:1:1 were end-capped with –COOH groups due to the excess BTA in synthesis. The other two synthesis ratios of 1:0:2 and 1:1:1 were end-capped with the –NH_2_ group because of the small proportion of BTA. Therefore, the strong signals at 9.35 ppm presented the phenyl rings between two –COOH groups of BPD-101 and BPD-211. The peak around 8.6 ppm (BPD-101 and BPD-211), 8.7 ppm (BPD-102) and 8.8 ppm (BPD-111) can be assigned to protons of the phenyl rings between the –COOH group and the –CONH group. The chemical shift at 7.7 ppm of BPD-102 and 8.2 ppm of BPD-111 were ascribed to the phenyl rings between two –CONH groups.

In addition, the chemical structures of the polyamide polymers and membrane were investigated with the FT-IR spectra, as shown in Figure 4a,b. The characteristic peak of the polyamide polymers at 1729 cm^−1^ is ascribed to the vibration of C=O and the peaks at 1490 and 1366 cm^−1^ (1506, 1404 in Figure 4b) belong to the vibration of N–H and C–N. The peaks at 3410 (Figure 4a) and 3444 cm^−1^ (Figure 4b) belong to the stretching vibration of O-H and the peaks at 1190 cm^−1^, 1020 cm^−1^ and 720 cm^−1^ (Figure 4a) are assigned to sulfonated groups of polymers. As a result, the four polymers are successfully synthesized. Figure 5 depicts the XRD patterns of BPD-*x* membranes. A typical peak (~20°) composed of crystalline regions of polyamide [58] clearly appears in all these patterns. Table 1 shows the molecular weight (Mw) and the polydisperisty index (PDI) of the polymers. All these results verify that BPD-*x* polymers are prepared successfully.

The BPD-*x* blend membranes can be used in the direct methanol fuel cell application. In this case, the low methanol permeability of the membranes is the key factor to prevent the fuel of methanol crossover from the anode to cathode. Figure 6a compares the methanol permeability through the BPD-*x* membrane and Nafion 117 membrane and gradually decreases as the monomer PPD content increases. The methanol permeability value of the BPD-*x* membranes was found to be in the range of 8.13 × 10^−7^ cm^2^/s to 17.40 × 10^−7^ cm^2^/s, much lower than the value through the Nafion 117 membrane. In particular, the methanol permeability value of the BPD-111 membrane is as low as 8.13 × 10^−7^ cm^2^/s, one order of magnitude smaller than that of the Nafion 117 membrane. The BPD-111 membrane containing the lowest methanol permeability of the BPD-*x* membranes was ascribed to the successful introduction of the non-sulfonated hydrophobic monomer PPD in synthesis and effective adjusting the hydrophobic groups of the sulfonic acid groups (–SO_3_H) and carboxylate groups (–COOH) in the BPD-*x* polymer. The flat and uniform topography SEM image of the cross-section surface of BPD-101 and BPD-211 membranes also illustrated the low methanol permeability, as shown in Figure 6b,c. The SEM images display a much smoother cross-section surface for the BPD-211 membrane than the BPD-101 membrane. In addition, the high degree of uniformity forms the basis for the stronger mechanical strength of the BPD-211 membrane compared with the BPD-101 membrane. The low methanol permeability of the BPD-*x* membrane indicates that the BPD-*x* proton exchange membrane is suitable for the application in DMFCs.

As one of the most important properties of a PEM, the proton conductivity of the membranes examined at different temperatures is displayed in Figure 7. The temperature-dependent proton conductivity of the BPD-*x* blend membranes is compared from 25 °C to 80 °C at 100% relative humidity. The proton conductivity increases with the increasing temperature for all the measured membranes due to the dependence of proton mobility on temperature. Unfortunately, the proton conductivity of the BPD-*x* membranes is much lower than that of Nafion 117 (80 °C, 0.19 S/cm). It can be found that the BPD-102 membrane shows the best proton-conductive performance among the four membranes. The tendency of proton conductivity in accordance with the measured IEC value of the membranes is listed in Table 2. The IEC value of the membranes rose from 0.14 to 0.36 mmol/g, which is also much lower than that of the Nafion 117 membrane. The noteworthy thing is that the conductivity is associated with the content of sulfonic acid (SC). Proton conductivity of the BPD-*x* membrane is low because of the low SC (Table 1). As compared in Figure 8, the AFM tapping phase image of the BPD-102 membrane with the best proton-conductive performance exhibits more remarkable hydrophilic/hydrophobic phase separation than BPD-111.

Water-retaining capacity of the PEM is one of the important factors because water serves as the proton carrier during proton transportation based on vehicle mechanisms [59,60], while excessive water would reduce the mechanical stability of the PEM [61]. The dimensional instability during the hydration/dehydration cycle for PEMFC operation could lead to interfacial delamination between the electrode and the PEM and thus significantly affect cell performance. The water uptake and volume swelling of the BPD-*x* membranes are shown in Table 2. The water uptake and volume swelling of the membrane prepared without the hydrophobic monomer PPD are higher than those of membranes with PPD monomer. The water uptake of membranes without PPD is 36.4% and 42.9%, but it is 10.7% and 18.6% with PPD monomer. The volume swelling reduces with decreasing water uptake. Therefore, the membrane fabricated by adjusting the non-sulfonated hydrophobic monomer PPD exhibits good dimensional stability.

The mechanical stability of the BPD-*x* proton-exchange membranes was subsequently measured for the consideration of long-term application in DMFC [62]. The tensile stress and elongation at break of the membranes are shown in Figure 9. According to the tensile stress and elongation at break bar graph (Figure 9), it is found that the tensile stress of the membranes increases with the ratio of the PPD monomer in synthesis. The tensile stress of the BPD-101 membrane is 7.26 MPa, the BPD-102 membrane is 6.18 MPa, the BPD-111 membrane is 15.54 MPa, and the BPD-211 membrane is 10.55 MPa. From the tensile stress graph (Figure 9), it was found that the non-sulfonated hydrophobic monomer p-phenylenediamine (PPD) was successfully introduced by adding different ratios of PDD in synthesis to adjust the sulfonic acid groups (–SO_3_H) and carboxylate group (–COOH) of the polymer. Therefore, the tensile stress was effectively improved by regulating the hydrophobic groups of the BPD-*x* polymer. The tensile stress of membranes is 7–15 MPa, which was lower than that of the Nafion 117 membrane (20 MPa). The BPD-111 membrane exhibits the best tensile stress of 15.54 MPa, which is a little lower than that of the Nafion 117 membrane and might be adequate for fuel cell-operation applications [63].

To evaluate the comprehensive performance of the proton-exchange membrane, the selectivity of the proton-exchange membrane represents the ratio of proton conductivity to methanol permeability [64]. According to the calculated formula, s = σ/p, the calculated results are listed in Table 3. It is clearly exhibited that the BPD-211 membrane displays by far the highest selectivity of the four membranes. The selectivity of the BPD-211 membrane develops a selectivity value of ca. 2.3 times higher than that of the BPD-111 membrane. From this result, we can see that the BPD-211 membrane is synthesized by introducing the non-sulfonated hydrophobic monomer p-phenylenediamine (PPD), which effectively regulates the hydrophobicity of the polymer. As a result, the BPD-211 membrane is considered as the optimal blend membrane for DMFC application.

As an important factor to evaluate the practicability of the PEM in fuel cells, the oxidative stability of the BPD-*x* blend membranes was studied. As shown in Figure 10, the retained weights of the BPD-*x* membranes treated in Fenton’s reagent (80 °C) for 1 h and 24 h were measured to be 90–97% and 89–93%, respectively. The BPD-111 membrane exhibited the highest oxidative stability among the four membranes. The measured residual weights of the BPD-111 membrane were 97% and 93% after treating in Fenton’s reagent (80 °C) for 1 h and 24 h, respectively. The measured value of the BPD-111 membrane is comparable to the value of the Nafion 117 membrane [29]. At the same time, this result indicated that the proton-exchange membrane is synthesized by introducing a non-sulfonated hydrophobic PPD monomer, which effectively enhances the oxidative stability of blend membrane.

The thermal stability of the BPD-*x* blend membranes was subsequently analyzed to assess the long-term application in DMFCs. The typical thermal stability of the BPD-x membranes was examined under a nitrogen atmosphere. The thermogravimetric analysis (TGA) curves of the BPD-*x* membrane are plotted in Figure 11. These BPD-*x* membranes show three-step weight-loss thermograms in the curve analysis. The first slight weight loss at about 200 °C is attributed to the evaporation of absorbed water and solvent. The second weight loss at about 500 °C is caused by the degradation of the sulfonic groups from the membrane. The degradation at about 650 °C is due to the decomposition of the main chains, include the breaking of the amide bond (–CO–NH–). And the 5% mass loss temperature of the membrane is higher than 200 °C. In conclusion, the BPD-*x* membranes exhibit satisfied thermal stability for the direct methanol fuel cell applications.

## 4. Conclusions

A sulfonated polyamide BPD-*x* was designed and successfully synthesized as a proton conductor. To enhance the tensile stress and to improve the dimensional stability of the proton-exchange membrane, PVDF was utilized as a binder for preparation of blend membranes with 50 wt.% binder to polymer ratios. Subsequently, the performances of these membranes as potential proton-exchange membrane materials for DMFC application were measured and compared with Nafion 117 membrane. The s-PA/PVDF membranes exhibited good methanol resistivity. The methanol permeability values of the membranes were found to be in the range of 8.13 × 10^−7^ cm^2^/s to 17.40 × 10^−7^ cm^2^/s. Remarkably, methanol permeability through the membrane is displayed to be much lower than that of Nafion 117, especially the methanol permeability value of BPD-111, which was as low as 8.13 × 10^−7^ cm^2^/s, one order of magnitude smaller than that of the Nafion 117 membrane. The BPD-111 membrane exhibited the best tensile stress of 15.54 MPa, which was very close to the Nafion 117 membrane and might be adequate for fuel cell-operation applications. These membranes also exhibited satisfied thermal stability for fuel cell applications, and the 5% mass loss temperature of the membrane is higher than 200 °C. Unfortunately, the proton conductivity of these membranes was lower than the Nafion 117 membrane. The tendency of proton conductivity is in accordance with the measured IEC value of the membranes. The IEC value of the membranes rises from 0.14 to 0.36 mmol/g, which is also much lower than that of the Nafion 117 membrane. The selectivity was comprehensively evaluated and the BPD-211 membrane is considered as the optimal membrane for DMFC application of these four blend membranes. Considering the good proton/methanol selectivity and mechanical/thermal stability, the fabricated BPD-*x* blend membranes show promising potential for application in PEM/DMFCs.

## Data Availability

The original contributions presented in the study are included in the article, further inquiries can be directed to the corresponding author.

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
