# Peer review of "A High-Methanol-Permeation Resistivity Polyamide-Based Proton Exchange Membrane Fabricated via a Hyperbranching Design"

_polymers, 2024, doi:10.3390/polym16172480_

Round 1
Reviewer 1 Report
Comments and Suggestions for Authors
The manuscript polymers-3113341 is devoted to the synthesis of membranes for methanol fuel cells. The relevance of the work is confirmed by the request for a transition to environmentally friendly energy sources. I believe that the work is novel (even in comparison with the previously published work of the authors) and can be published after significant revision. My comments and suggestions are listed below.
The last paragraph of the introduction needs to be seriously revised. Firstly, the authors should draw the boundaries more clearly and compare the results obtained with the results obtained in their work 10.3390/polym10101175. Of course, the synthesis conditions, monomers and resulting membranes are different, but the authors should show more clearly how this differs from the previous series of experiments. Secondly, I suggest you consider rephrasing the title so that readers don't confuse your work in the future. For example, you can definitely get rid of the term “Optimization”. This term would be appropriate if you carried out a second series of experiments with the selection of membrane composition based on the results of characterization of the first series. Matching the titles of your articles may lead to avoidance of citing your works simultaneously in one article on proton exchange membrane by other authors.
The last paragraph of the introduction should be shortened. Leave the conclusions and findings for the relevant parts of the article. Don't list it here. This should be the purpose of your new work, its main differences from other works (and yours in particular), in other words, the rationale for this article.
Avoid wording similar to your previous article. Again, this reduces the value of your new work. Paraphrase the names of the figures, tables, and also the text in some paragraphs where you used the same figures of speech.
What determines the concentrations, time and temperature in tests with Fenton's reagent? In the literature there are different quantitative compositions, different temperature ranges, different sizes of pieces of membrane. The dimensions of pieces of membrane should be indicated.
The distance between the electrodes and width of cell for measuring proton conductivity, as well as thickness of membranes should be specified. How was the thickness of the membranes controlled? Was it set constant for all samples? Describe a solution for measuring proton conductivity. How was the conductivity of the solution taken into account? Are the Pt electrodes (Fig. 1) for signal measurement located on one side of the membrane? How does such a measurement scheme make it possible to obtain the membrane signal?
Similar notes for measuring diffusion permeability. It is required to indicate the active area of the membranes and the volume of the feed and receiving solution. What determines the choice of concentration of methanol solution?
The membrane images in Figures 5b and 5c are not very informative. Please fix their quality.
The reasons for the low proton conductivity of the resulting membranes are not clear (lines 263-268). Please clarify. It is necessary to analyze the literature and compare the results with membranes obtained under similar conditions.
Table 1: I consider the accuracy of the indicated values to be unnecessary. Authors should round values and also indicate confidence intervals
The BPD-211 membrane is recognized as optimal. However, according to Table 1, this membrane does not have the highest exchange capacity. In addition, the value of its exchange capacity is several times less than the similar value for the Nafion membrane. Do you think this membrane can compete with the Nafion membrane? Perhaps, in addition to the described stationary parameters of membranes, it is necessary to evaluate the parameters of the process using membranes? For example, performance and energy consumption. Have membranes been tested in methanol fuel cells?
Comments on the Quality of English Languageno specific comments
Author Response
I would like to resubmit the revised manuscript. We greatly appreciate the constructive comments and suggestions and have revised the manuscript accordingly. All revisions are marked in yellow in the manuscript.
Sincerely yours,
Liying Ma

Reviewer 2 Report
Comments and Suggestions for Authors
The authors have prepared a set of sulfonated polyamide polymers and made membranes by blending the polymers with PVDF. They characterized the membrane properties relevant for application in DMFCs. The paper continues a large number of similar works by the authors and by others, as appropriately cited in the manuscript.
The novelty is therefore rather limited and it is of interest only for readers working on similar polymers. It can be published with minor revisions:
1. By changing the composition of the three monomers, the authors change hydrophobcity, ion exchange capacity and cross-linking. This should be discussed more clearly, the discussion currently focusses almost only on hydrophobicity.
2. Please discus the effect of membrane thickness and give the thickness of the memranes prepared in this work. Higher selectivity means that there is a membrane thickness, where area resistance is the same as for Nafion 117, while methanol permeation is lower, what would be this thickness? What would the mechanical properties be for such a membrane?
3. Have the mechanical properties measured on dry membranes? What would the effect of water uptake be?
4. Reference [1] is inappropriate to support the first sentence. I am sure plenty of publications, for example by the WHO exist to support this better.
5. A number of small mistakes need to be corrected:
line 32: proton exchange (not polymer exchange)
line 75: mechanical strength (not property)
line 111 ff: x = 2 mmol (not 2x mmol)
Scheme 1: here, DSA is y and PPD is z, opposite to the text, please make sure this is always the same
line 124: please give the amounts of polymers and solvents as well as the resulting membrane thickness
line 241: the maximum value for methanol permeability is around 18*10^-7 cm²/s (not 10.5)
line 267: IEC is lower than that of Nafion 117: what can be done to achieve a higher IEC?
line 351: various binders to polymer ratios: this is wrong, this was done in a previous paper, but not in this, where it is always 50% PVDF as stated above.
Comments on the Quality of English Language
please check especially singular/plural
Author Response

(The authors gave the same response as above.)

Reviewer 3 Report
Comments and Suggestions for Authors
The authors must perform additional experiments for considering this manuscript to get published in the Polymers.
1. Provide the purity of all the chemicals. Provide the Mw of all the polymers.
2. Include the thickness of all the membranes.
3. Calculate the degree of sulfonation to all the materials. DS% is an important factor in the membrane.
4. the DMFC performance must included
5. Through plane conductivity must be measured.
6. Membrane cross- sectional analysis must be carried out.
7. AFM results must be include to understand the hydrophilicity in the membrane.
8. The TGA degradation must be explained based on the Polymers decomposition
9. In Table 2, include the results of Nafion 117
10. In Table 3, include the results of Nafion 117
11. Provide the XRD and FTIR for all the composite materials.
12. Figure 6 includes the results of Nafion 117
13. Figure 7, include the results of PVDF and Nafion 117
14. Figure 8, include the results of PVDF and Nafion 117
Comments on the Quality of English LanguageModerate editing of English language required
Author Response

(The authors gave the same response as above.)

Round 2
Reviewer 1 Report
Comments and Suggestions for Authors
Dear authors, thank you for the corrections. However, I have some small comments left. I still do not understand Figure 1. Both the polarizing and measuring electrodes are located on one side of the membrane according to the figure 1. There should be one measuring and one polarizing electrode on each side, as far as I understand. Please reconsider
Comments on the Quality of English Languageno specific comments
Author Response
Comments 1: Dear authors, thank you for the corrections. Howerer, I have some small comments left. I still do not understand Figure 1. Both the polarizing and measuring electrodes are located on one side of the membtane according to the Figure 1. There should be one measuring and one polarizing electrode on each side, as far as I understand. Please reconsider.
Response 1: Thank you for pointing this out. I agree with this comment. Therefore, Figure 1 has been modified in manuscript.

Reviewer 2 Report
Comments and Suggestions for Authors
The concerns raised in my previous review have been appropriately addressed. The paper can be published in the current form.
Author Response
Commetns 1: The concerns raised in my previous review have been appropriately addressed. The paper can be pubished in the current form.
Response 1: Thank you for your comments.

Reviewer 3 Report
Comments and Suggestions for Authors
Accept
Comments on the Quality of English LanguageExtensive editing of English language required
Author Response
Comments 1: Comments on the Quality of English Language Extensive editing of English language required.
Response 1: Thank you for pointing this out. I agree with this comment, and I have edited English language and marked in yellow in the manuscript.
